# A Multi-Gene Signature of Non-Muscle-Invasive Bladder Cancer Identifies Patients Who Respond to Immunotherapies Including Bacillus Calmette–Guérin and Immune Checkpoint Inhibitors

**DOI:** 10.3390/ijms25073800

**Published:** 2024-03-28

**Authors:** Seung-Woo Baek, Sun-Hee Leem

**Affiliations:** 1Metabolic Regulation Research Center, Korea Research Institute of Bioscience and Biotechnology (KRIBB), Daejeon 34141, Republic of Korea; baek@kribb.re.kr; 2Department of Biomedical Sciences, Dong-A University, Busan 49315, Republic of Korea; 3Department of Health Sciences, The Graduate School of Dong-A University, Busan 49315, Republic of Korea

**Keywords:** high-risk non-muscle-invasive bladder cancer, the multi-gene signature, molecular subtype, prognosis, BCG shortage, immunotherapy

## Abstract

Approximately 75% of bladder cancer cases originate as non-muscle-invasive bladder cancer (NMIBC). Despite initial diagnosis, NMIBC commonly recurs, with up to 45% advancing to muscle-invasive bladder cancer (MIBC) and metastatic disease. Treatment for high-risk NMIBC typically includes procedures like transurethral resection and, depending on recurrence risk, intravesical chemotherapy or immunotherapy such as Bacillus Calmette–Guérin (BCG). However, persistent shortages of BCG necessitate alternative first-line treatments. We aim to use a multi-gene signature in high-risk NMIBC patients to determine whether patients may benefit from immune checkpoint inhibitors (ICIs) as an alternative to BCG and to evaluate their clinical utility. The multi-gene signature obtained from the three independent NMIBC cohorts was applied to stratify the UROMOL2016 cohort (*n* = 476) using consensus clustering. Each subtype was distinguished by biological pathway analysis. Validation analysis using a machine learning algorithm was performed in six independent cohorts including the BRS (*n* = 283) cohort treated with BCG and the IMvigor210 (*n* = 298) clinical trials treated with PD-L1 inhibitors. Based on consensus cluster analysis, NMIBC patients in the UROMOL2016 cohort were classified into three classes exhibiting distinguished characteristics, including DNA damage repair (DDR). Survival analysis showed that the NMIBC-DDR class had the highest rates of disease progression (progression-free survival, *p* = 0.002 by log-rank test) in the UROMOL cohort and benefited from BCG and ICIs (respectively, *p* = 0.02 and *p* = 0.03 by log-rank test). This study suggests that the multi-gene signature may have a role in identifying high-risk NMIBC patients and improving the responsiveness of ICIs. Additionally, we propose immunotherapy as a new first-line treatment for patients with high-risk NMIBC because of the shortage of BCG supply. It is important to help more patients prioritize cancer immunotherapy.

## 1. Introduction

Bladder cancer accounts for 83,190 cancer-related deaths annually in the United States [1]. Approximately 75% of BC patients are initially diagnosed with non-muscle-invasive bladder cancer (NMIBC). Although NMIBC patients generally have a favorable prognosis, with a five-year survival rate of approximately 85%, the recurrence rate is approximately 70%, and up to 45% of these cases progress to muscle-invasive bladder cancer (MIBC) and metastatic disease [2].

According to the 2023 European Association of Urology (EAU) treatment guidelines for NMIBC, all patients undergo transurethral resection of bladder tumor (TURBT) and subsequent treatment is determined by the EAU or European Organization for Research and Treatment of Cancer (EORTC) risk scores. Patients with low scores are offered intravesical chemotherapy, patients with intermediate scores are offered Bacillus Calmette–Guérin (BCG) or chemotherapy, and patients with high scores are offered intravesical BCG treatment or radical cystectomy (RC) [3].

BCG treatment is the optimal adjuvant therapy for high-risk NMIBC patients after TURBT, but this treatment has several limitations. Patients may experience relapses at rates of up to 40% at 2 years, while other vulnerable populations, such as elderly and immunocompromised individuals, may have reduced immunological responses [4]. However, the most important issue with BCG is ongoing production shortages, which have left many urological practices in need of alternative first-line treatments that are effective and readily available [5,6]. Additionally, in recent years, due to the impact of the pandemic, the BCG supply has been disrupted worldwide, increasing the urgency of developing first- or second-line treatments for NMIBC [7]. From this perspective, immunotherapy treatment may be a good alternative to overcome the BCG shortage. New strategies involving immunotherapy are currently being evaluated in several clinical trials [8]. These efforts suggest that preferential immunotherapy for high-risk NMIBC patients may reduce cancer progression and recurrence.

In this study, we used transcriptomic data from NMIBC patients to construct a multi-gene signature to identify three molecularly distinct subtypes. In addition, through a machine learning-based prediction model, we performed a validation analysis on six independent bladder cancer patient cohorts, including a cohort with BCG and immune checkpoint inhibitor (ICI) records, and estimated the prognostic relevance.

## 2. Results

### 2.1. Identification of Distinct Molecular Subtypes of NMIBC

The baseline characteristics of the 1996 NMIBC and MIBC patients and the schematic workflow of this study are detailed in Figure 1 and Table 1.

We used the top 5000 genes based on standard deviation in the three NMIBC cohorts to construct a common 1528 gene signature. Based on this signature, a consensus clustering method was used to determine the optimal number of subgroups for distinguishing NMIBC patients in the UROMOL2016 cohort, which was best when patients were divided into three subgroups (Appendix A).

To determine the activation of biological pathways in each subgroup, Kyoto Encyclopedia of Genes and Genomes (KEGG) pathway analysis of the Database for Annotation, Visualization, and Integrated Discovery (DAVID) was applied to the genes (FDR < 0.25). Cluster 1 was enriched in cell cycle and DNA damage and repair (DDR)-related genes, which included the Fanconi anemia pathway and homologous recombination pathways (named NMIBC-DDR). Cluster 2 was enriched in epithelial–mesenchymal transition (EMT) and T cell activation-related genes, which included those involved in focal adhesion, the PI3K-AKT signaling pathway, the regulation of the actin cytoskeleton pathway, and antigen processing and presentation-related genes (named NMIBC-ET). Cluster 3 was enriched in lipid metabolism and ABC transporters-related genes. And this class also enriched TGF-beta, MAPK, and Hippo signaling pathways (named NMIBC-F3M, Appendix A).

### 2.2. Characteristics of Distinct Molecular Subtypes of NMIBC

An integrated analysis was performed to evaluate the prognostic impact of each subgroup of NMIBC patients. We selected seven genes representing each subgroup and collected clinical information including stage, grade, the taxonomy of the UROMOL consortium, risk scores, and various signatures (Figure 2A). After exploring NMIBC-DDR, we found that DNA damage response-related genes, including *AURKA*, *CENPF*, *CHEK1*, *EXO1*, *FANCB*, *FOXM1*, and *TOP2A*, were enriched in a large number of patients with T1-high grade NMIBC (59 out of 78, 76%). The EAU and EORTC risk scores were also high (respectively, 81 out of 203, 40%, and 100 out of 174, 57%), high grade and T2-4 patients (respectively, 115 out of 192, 60%, and 7 out of 16, 44%), and the late cell cycle and DNA replication signatures was highly expressed in NMIBC-DDR. Exploring NMIBC-ET, we found that it was enriched in EMT and T cell activation-related genes, including *COL8A1*, *DCN*, *LUM*, *PDGFRA*, *PTGS1*, *VCAM1*, and *VIM*. Patients mainly included Class 2b (90 out of 160, 56%) and showed high expression of EMT, hypoxia, CD8 T cell, and Interferon gamma signatures. In NMIBC-F3M, the expression of the *FGFR3*, *FOXQ1*, *GATA3*, *HOXA1*, *KRT13*, *MOGAT2*, and *PLCD3* genes was identified. The patients were mainly classified as Class 1 and Class 3 (respectively, 65 out of 99, 65%, and 86 out of 96, 89%) and exhibited high expression of the early cell cycle and FGFR3 signatures (Figure 2A,B). The progression rate of NMIBC-DDR was significantly higher than that of the remaining classes (log-rank test, *p* = 0.002, Figure 2C).

### 2.3. Prognostic Impact of Distinct Molecular Subtypes of NMIBC

We used multi-gene signatures that distinctly identified three subtypes of NMIBC and we applied a machine learning method (random forest) to train a prediction model based on the 1528 gene signature to evaluate the prognostic relevance. The transcriptomic data from 393 randomly selected NMIBC patients (80%) in total (training dataset) were pooled to form a prediction model able to estimate the probability that a particular NMIBC patient belonged to one of the three subtypes. To prevent overfitting in the prediction model, we selected appropriate hyper-parameters through grid search (Appendix A) and retrained the model with the remaining NMIBC patients (20%) in total (test dataset; accuracy 94%, Appendix A and Figure 3).

When this prediction model was applied to the validation cohorts, three classes were clearly identified and prognostic relevance was determined (Appendix A). Consistent with the previous results in UROMOL2016, most high-risk NMIBC patients were included in the NMIBC-DDR class (respectively, *p* = 0.002, *p* < 0.001, and *p* = 0.003 by log-rank test, Figure 4A–C).

### 2.4. NMIBC-DDR Was Associated with a Better Response to Immunotherapy Such as BCG and Immune Checkpoint Inhibitor Treatment

In the BRS cohort with BCG treatment, three distinct classes were identified using the prediction model, and the lowest rates of disease progression were observed in NMIBC-DDR (*p* = 0.02 by log-rank test, Figure 4D,E). In addition, in the cases of BCG response rate, the trend was confirmed to be highest in NMIBC-DDR and lowest in NMIBC-ET (Figure 4F). In the classification of BRS, the BRS3 subtype was mostly included in NMIBC-ET. To confirm whether the new NMIBC classification system consistently had prognostic relevance in the MIBC cohort, validation analysis was performed on MIBC patients from The Cancer Genome Atlas (TCGA) cohort and the immunotherapy clinical trial cohort (IMvigor210 trial). The data were analyzed separately by a prediction model and three classes were assigned (Appendix A). For the TCGA cohort, a prognostic relevance could be found for the three classes (*p* = 0.03 by log-rank test, Figure 5A). In the case of the IMvigor210 trial, three distinct classes and prognostic relevance were identified. NMIBC-DDR, which had the highest rate of progression in NMIBC and poor prognosis for MIBC patients, showed the most improved prognosis in the IMvigor210 trial (*p* = 0.03 by log-rank test, Figure 5B,C). When the objective response rate (ORR) was compared among these classes, NMIBC-DDR exhibited a higher response rate than the other subtypes (chi-square test, *p* < 0.001, Figure 5D).

### 2.5. Comparison of Prognostic Relevance of Clinical Risk Factors and Molecular Subtypes of NMIBC

To confirm the prognostic independence of a multi-gene signature-based prediction model for bladder cancer cohorts, we applied Cox proportional hazards regression analyses to the three classes in the independent cohorts.

Univariate analysis confirmed various factors, such as stage, grade, risk scores, and taxonomy, as significant indicators of disease progression in NMIBC (Table 2). Additionally, the multi-gene signature, PD-L1 expression in immune cells, and the tumor mutation burden (TMB), were identified as major indicators of responsiveness to ICI treatment (Table 2). Multivariate analysis with the EORTC risk score identified that the multi-gene signature remained a statistically significant indicator for disease progression in the UROMOL cohort (hazard ratio, 3.48; 95% confidence interval, 1.46–8.29; *p* = 0.004; Table 2). Multivariate analysis with the EORTC risk score identified the multi-gene signature as a statistically significant indicator of a trend toward reduced disease progression with BCG response in the BRS cohort (hazard ratio, 0.62; 95% confidence interval, 0.04–0.95; *p* = 0.03; Table 2). In addition, the multi-gene signature still remained an associated indicator of responsiveness to ICI treatment in metastatic bladder cancer patients (hazard ratio, 0.69; 95% confidence interval, 0.51–0.94; *p* = 0.018; Table 2). In the GSE163029, GSE32894, GSE13507, and TCGA cohorts, the multi-gene signature remained a significant indicator through univariate analysis, but only reached statistical significance in multivariate analysis in the GSE13507 and TCGA cohorts (Appendix A).

## 3. Discussion

This study demonstrated that the multi-gene signature can stratify NMIBC patients into three molecularly distinct subtypes and predict patient prognosis. The ability of this multi-gene signature to predict patient prognosis was validated in independent bladder cancer cohorts. Furthermore, we identified potential benefits of immunotherapy such as BCG and ICIs in NMIBC-DDR. Given the growing urgency of developing first- or second-line treatments for high-risk NMIBC patients due to the global BCG shortage [5,6,7,9], and the view that immune checkpoint inhibitor-based treatment may be an alternative for these patients [8], the multi-gene signature is valuable as an important indicator.

The genes in each cluster were assigned to three classes (NMIBC-DDR, NMIBC-F3M, and NMIBC-ET) and identified as molecularly distinct through biological pathway analysis. The risk score of the NMIBC-F3M class was also low in the EAU and EORTC risk scores, and a relative increase in the early cell cycle was observed. In addition, there was a relative increase in FGFR3 signature, a higher proportion of patients with Class 1 and 3 in the UROMOL2021 taxonomy. The NMIBC-ET class had relatively high EMT and hypoxia signatures, and at the same time, activation of signatures associated with T cell activation was also observed. It is known that the EMT in cancer is the cause of acquiring resistance to many other drugs. In particular, it changes the tumor microenvironment (TME), making it difficult for immune cells to reach it, and metastasis can become possible through cell remodeling [10]. These characteristics are significantly relevant in several studies showing that T cells are activated but suppress the effect of immunotherapy at the TME such as cancer-associated fibroblast (CAF) [11]. The low response rate to immunotherapy such as BCG or ICIs was associated with those observed in the NMIBC-ET class of the BRS and IMvigor210 cohorts (Figure 4E,F and Figure 5B,D). The NMIBC-DDR class had a high rate of disease progression and included many T1-risk patients such as those with T1-high grade and high EAU or EORTC risk scores.

Clearly, in the three independent NMIBC cohorts (GSE163209, GSE32894, and GSE13507), patients classified as NMIBC-DDR had a higher rate of cancer progression, whereas slowed cancer progression was observed in the BRS cohort treated with BCG. Similarly, in MIBC, patients classified as NMIBC-DDR in the TCGA cohort had a poor prognosis, but the prognosis was confirmed to be improved in the IMvigor210 trial with ICI treatment.

PD-L1 is expressed on tumor cells and peripheral immune cells and is an important part of tumor immune evasion mechanisms. Many follow-up studies have revealed that PD-L1 expression in immune cells is a major target for immunotherapy [12,13]. PD-L1 expression, especially on CD8-T cells, may be involved in tumor evasion of immune responses. Excessive PD-L1 expression is associated with impaired function of CD8-T cells, which may have negative effects on tumor invasion and metastasis [14]. Activation of the cell cycle and DDR pathways in cancer cells ultimately increases the tumor mutational burden (TMB), making them targets of immune cells. The TMB causes a greater number of mutations in cancer cells, causing them to become neoantigens which are modified proteins that do not exist in normal cells. Neoantigens are recognized as foreign substances and serve as targets for the immune system. The higher the TMB, the more diverse neoantigens are produced, making the tumor more visible to the immune system [15]. Therefore, the result that NMIBC-DDR patients show potential benefits from immunotherapy such as BCG or ICIs can be interpreted as a result that well reflects these characteristics (Figure 4E,F and Figure 5B,D). Because the BRS cohort mainly included patients with high-grade NMIBC (94.7%, Table 1) and the IMvigor210 cohort was a clinical trial conducted on patients with metastatic muscle-invasive bladder cancer [16], in this study the progression-free survival of the BRS cohort or the overall survival of the IMvigor210 cohort of MIBC may show a much worse trend than other compared cohorts.

Taken together, high-risk NMIBC patients, such as those with high EAU or EORTC risk scores, are more likely to experience frequent cancer recurrence and progression. BCG treatment and cystectomy should be discussed in these patients, according to the EAU treatment guidelines. In the case of NMIBC-DDR identified by the multi-gene signature, patients could potentially be classified as high-risk NMIBC patients, and a favorable outcome was also observed in BCG response. In addition, MIBC patients corresponding to NMIBC-DDR showed a poor prognosis, but the results of the IMvigor210 clinical trial suggest that these patients may have a favorable ICI response, as observed in a favorable BCG treatment response. Considering the global shortage of BCG supply and the misfortune of the poor quality of life for patients after cystectomy, immunotherapy based on immune checkpoint inhibitors is likely to be a good alternative as first or second treatment for high-risk NMIBC patients (Figure 6).

The results of this study have several limitations. First, in vitro or in vivo experiments should be further discussed to confirm and verify the research results. Additionally, the continuity of results must be evaluated through new bladder cancer immunotherapy clinical trial data that will be added in the future. In conclusion, we have confirmed the value of the multi-gene signature in distinctly identifying NMIBC and observed a potential benefit for immunotherapy. Due to the shortage of BCG supply and the quality of life of patients, immunotherapy based on immune checkpoint inhibitors may be proposed as a treatment for high-risk NMIBC patients.

## 4. Materials and Methods

### 4.1. Public Datasets of NMIBC and MIBC Patients

We obtained clinical information and gene expression data from NMIBC and MIBC patients from public databases. The raw data from the UROMOL consortium were downloaded from the European Genome-Phenome Archive (EGA) under accession numbers EGAS00001001236 (UROMOL2016) and EGAS0001004693 (UROMOL2021) [17,18]. Prior to the analysis, we performed the analysis with version UROMOL2016, which included 16 MIBC patients, and used version UROMOL2021 for clinical information. Gene expression data for the bladder cancer microarray study were downloaded from the Gene Expression Omnibus (GEO) database from the National Center for Biotechnology Information (NCBI) under accession numbers GSE32894, GSE163209, and GSE13507 [19,20,21]. The BRS cohort data were obtained through in the paper [22]. The TCGA data were obtained through the cancer browser (https://xenabrowser.net, accessed on 30 January 2020). The raw data for the IMvigor210 trial were downloaded from the EGA under accession numbers EGAS00001002556 (IMvigor210 trials) [16].

### 4.2. Processing of Transcriptomic Data and Clustering Analysis

Reference genome sequence data from *Homo sapiens* were obtained from the Ensemble genome browser (assembly ID: GRCh38). Reference genome indexing and read mapping of samples were performed using STAR software (ver. 2.6.1a) [23]. FeatureCounts (ver. 1.6.2) software was then used to calculate generated binary alignment map files [24]. All gene expression data were separately log2 transformed and quantile normalized. For RNA-seq data, the read counts per million fragments mapped (CPM) of each sample were used to estimate the expression level of each gene. Genes with an average CPM value less than 1 for the entire sample were removed. Regardless of the data type (RNA-seq or microarray), genes with expression levels that differed more than twice the median in more than 10% of the samples were used for the main analysis. ConsensusClusterPlus (R package, ver. 1.52.0) analysis was performed to determine the appropriate number of clusters for NMIBC patients [25].

### 4.3. Development of Prediction Model Based on the Multi-Gene Signature

We identified the top 5000 genes based on standard deviation in three independent NMIBC patient cohorts (UROMOL2016, BRS, and GSE163209), and selected 1528 commonly included genes. The process of selecting common genes from three independent NMIBC patient cohorts involved selecting as many common features as possible, taking molecular heterogeneity into account. Consensus clustering analysis was performed using the multi-gene signature. The optimal k value (k = 3) was chosen considering both the delta area and the cumulative density function (CDF) plot. To validate the prognostic relevance of the subtypes derived from UROMOL2016, we trained a prediction model using a random forest machine learning algorithm (R package, randomForest ver. 4.7–1.1) and subsequently improved performance by modifying hyper-parameters (m_try_, node size, and sample size) via grid search. The m_try_ helps balance reasonable prediction strength. Node size is the most common hyper-parameter that controls tree complexity. If the nodes are too large (i.e., increasing tree depth and complexity), overfitting problems may occur. If the training dataset is too small, the performance of the prediction model may deteriorate due to insufficient learning. Gene expression data used to build the prediction model were preprocessed through normalization and standardization.

### 4.4. Biological Pathway and Statistical Analysis

To explore significantly enriched functions, KEGG pathway analysis was performed using the DAVID tool (https://david.ncifcrf.gov) accessed on 1 February 2024 with significance criteria (FDR < 0.25). Kaplan–Meier plots were used to estimate associations between subgroups and survival outcomes by the log-rank test. Comparisons of objective responses were estimated using the chi-square test. All statistical analyses were performed in the R 4.2.1 language environment (https://www.r-project.org) accessed on 22 June 2022.

### 4.5. Data Availability

The gene expression data and clinical information are available in the GEO public database (https://www.ncbi.nlm.nih.gov/geo) (accessed on 1 January 2024) under accession numbers GSE32894, GSE163209, and GSE13507. The gene expression data and clinical information of TCGA are available in the cancer browser (https://xenabrowser.net) accessed on 30 January 2020. The gene expression data and clinical information of BRS are available in the Appendix A of this paper. The FASTq files and clinical information are available in the EGA public database (https://web2.ega-archive.org/) accessed on 1 June 2023 under accession numbers EGAS00001001236 (UROMOL2016), EGAS0001004693 (UROMOL2021), and EGAS00001002556 (IMvigor210 trial).

## 5. Conclusions

This study confirmed the utility of a multi-gene signature in stratifying non-muscle-invasive bladder cancer (NMIBC) patients into three molecularly distinct subtypes and predicting their prognosis. It demonstrated that the multi-gene signature effectively predicts patient outcomes and its relevance was validated across independent bladder cancer cohorts. This study also highlighted the potential advantages of immunotherapy, such as Bacillus Calmette–Guérin (BCG) and Immune Checkpoint Inhibitors (ICIs), especially for the NMIBC subtype associated with DNA damage response (NMIBC-DDR), amidst the global shortage of BCG and the search for alternative treatments for high-risk NMIBC patients. The findings suggest that patients in the NMIBC-DDR class, characterized by high rates of disease progression and poor prognosis in standard settings, may benefit from immunotherapy due to an increased tumor mutational burden (TMB), which enhances the visibility of tumors to the immune system through neoantigens. Consequently, this study supports immunotherapy with ICIs as a viable first- or second-line treatment option for high-risk NMIBC patients, offering a potential improvement in patient outcomes and quality of life while addressing the limitations of current treatment modalities due to BCG supply issues.

## Figures and Tables

**Figure 1 ijms-25-03800-f001:**
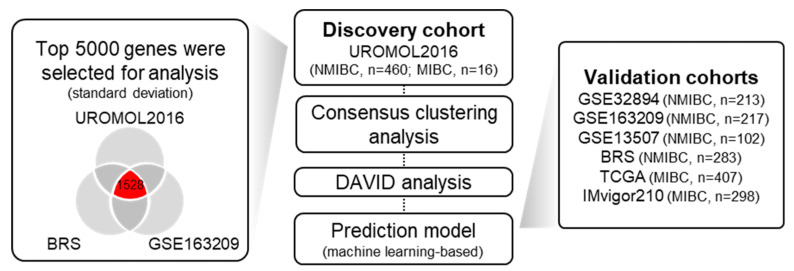
The workflow diagram for this study. First, we identified the top 5000 genes based on standard deviation in three independent non-muscle-invasive bladder cancer (NMIBC) cohorts and selected 1528 commonly included genes. Next, consensus clustering analysis was performed to determine the appropriate number of clusters. Then, Database for Annotation, Visualization, and Integrated Discovery (DAVID) analysis was performed to explore the significantly enriched pathways. Finally, we constructed a machine learning-based prediction model and verified the prognostic relevance. NMIBC, non-muscle-invasive bladder cancer; MIBC, muscle-invasive bladder cancer.

**Figure 2 ijms-25-03800-f002:**
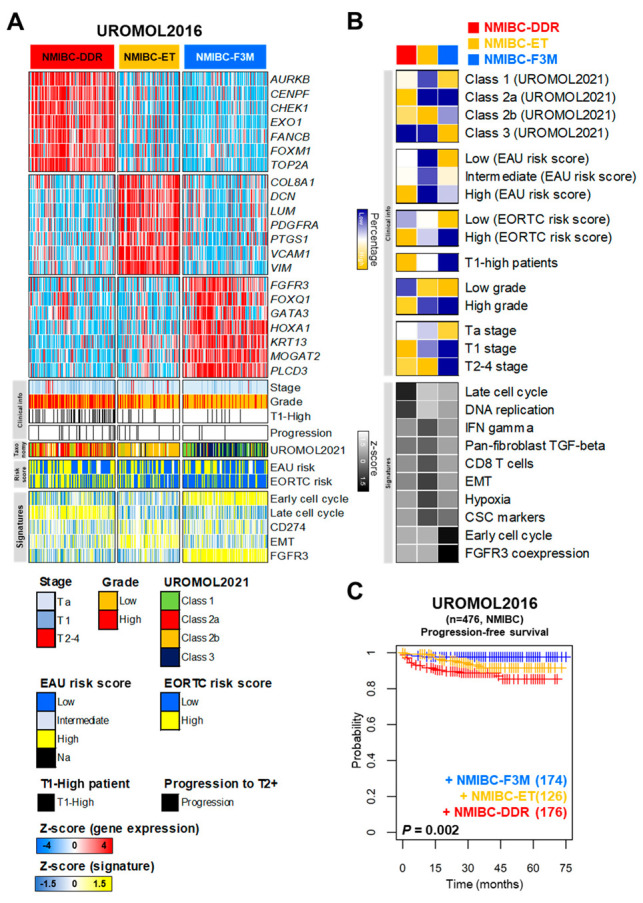
Characteristics of three classes categorized by the multi-gene signatures and estimation of their prognostic relevance in NMIBC patients. (**A**) The heatmaps, clinical information, taxonomies, risk scores, and signatures of UROMOL2016 (*n* = 476) were grouped into three classes according to the multi-gene signature. Seven genes representing each class were selected. The colors in the heatmaps reflect relatively high (red) and low (blue) (**B**) distributions of taxonomy, clinical information, risk scores, and signatures according to the three classes. (**C**) Kaplan–Meier plot of the three classes in UROMOL2016 (progression-free survival, *p* = 0.002 by the log-rank test).

**Figure 3 ijms-25-03800-f003:**
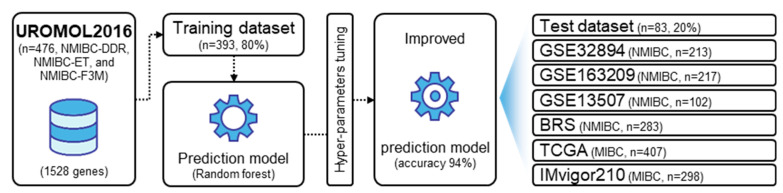
Development of a prediction model based on the multi-gene signature. The random forest algorithm was used. A prediction model was built based on transcriptome data from 393 randomly selected patients (80%, training dataset) among all patients. To prevent overfitting of the prediction model, we selected hyper-parameters through grid search and retrained the prediction model.

**Figure 4 ijms-25-03800-f004:**
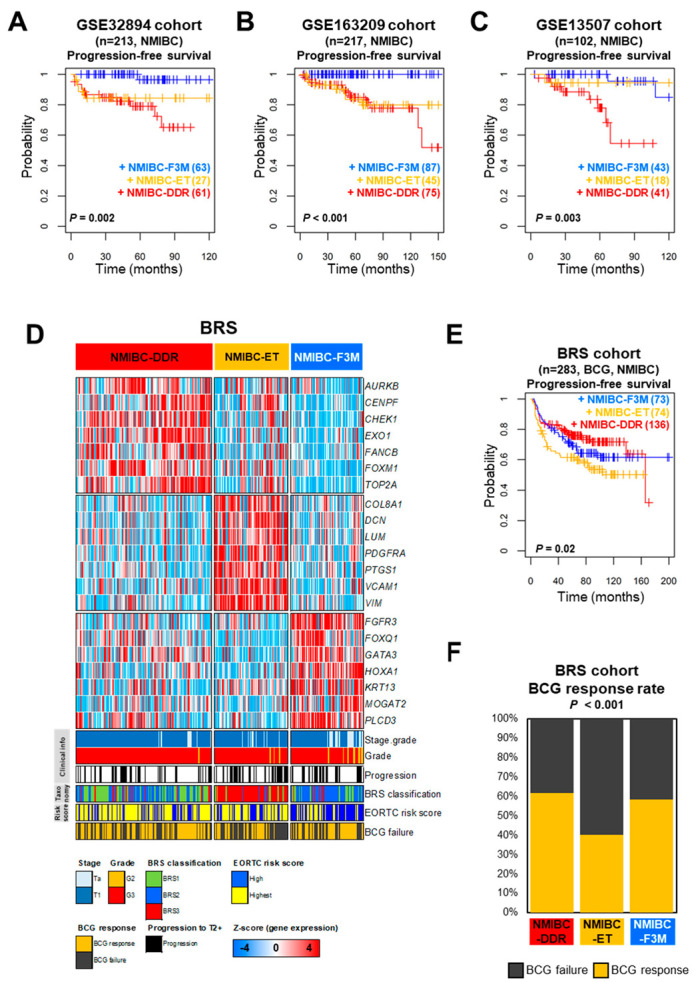
Estimation of prognostic relevance based on prediction model. (**A**) Kaplan–Meier plot of three classes in GSE32894 (progression-free survival, *p* = 0.002 by the log-rank test). (**B**) Kaplan–Meier plot of three classes in the GSE163209 cohort (progression-free survival, *p* < 0.001 by the log-rank test). (**C**) Kaplan–Meier plot of three classes in the GSE13507 cohort (progression-free survival, *p* = 0.003 by the log-rank test). (**D**) The heatmap and clinical information of BRS (*n* = 283) were divided into three classes according to the prediction model. Seven genes representing each class were selected with colors in the heatmap reflecting relatively high (red) and low (blue) levels. (**E**) Kaplan–Meier plot of three classes in the BRS cohort (progression-free survival, *p* = 0.02 by the log-rank test). (**F**) The bar chart represents the response rate for BCG treatment in each of the three classes. The *p*-values were calculated using the chi-square test.

**Figure 5 ijms-25-03800-f005:**
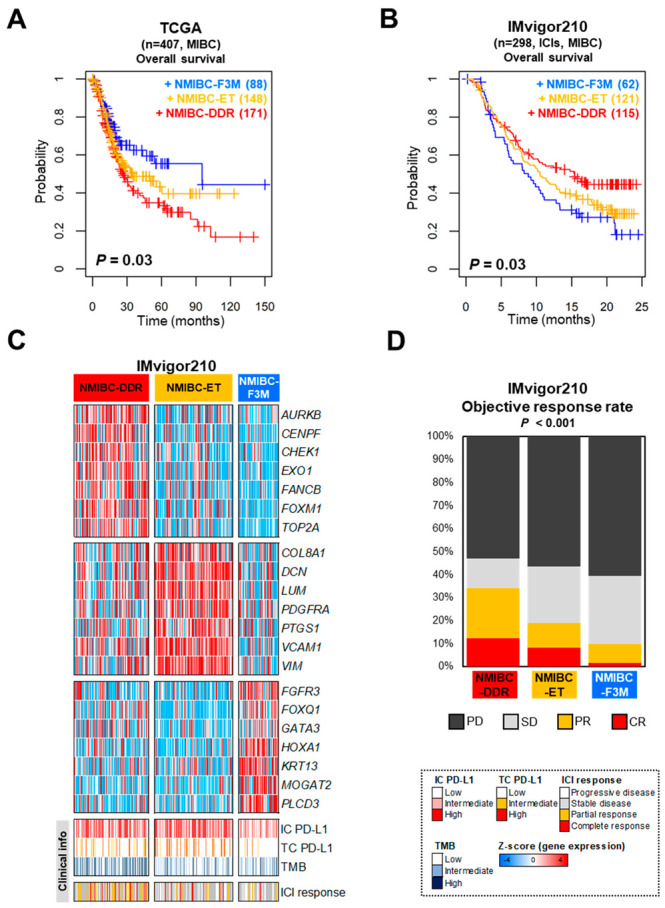
Characteristics of the three classes categorized by the multi-gene signatures and estimation of their prognostic relevance in MIBC patients. (**A**) Kaplan–Meier plot of the three classes in TCGA showed a significant difference in overall survival (*p* = 0.03 by the log-rank test). (**B**) Kaplan–Meier plot of the three classes in IMvigor210 with ICI treatment showed a significant difference in overall survival (*p* = 0.03 by the log-rank test). (**C**) The heatmaps and clinical information of IMvigor210 (*n* = 298) were divided into three classes according to the prediction model. Seven genes were selected representing each class, with colors in the heatmap reflecting relatively high (red) and low (blue) levels. IC PD-L1, PD-L1 expression in immune cells; TC PD-L1, PD-L1 expression in tumor cells; TMB, tumor mutation burden. (**D**) The bar chart represents the objective response rate by immunotherapy in each of the three classes. The *p*-values were calculated using the chi-square test. PD, progressive disease; SD, stable disease; PR, partial response; CR, complete response.

**Figure 6 ijms-25-03800-f006:**
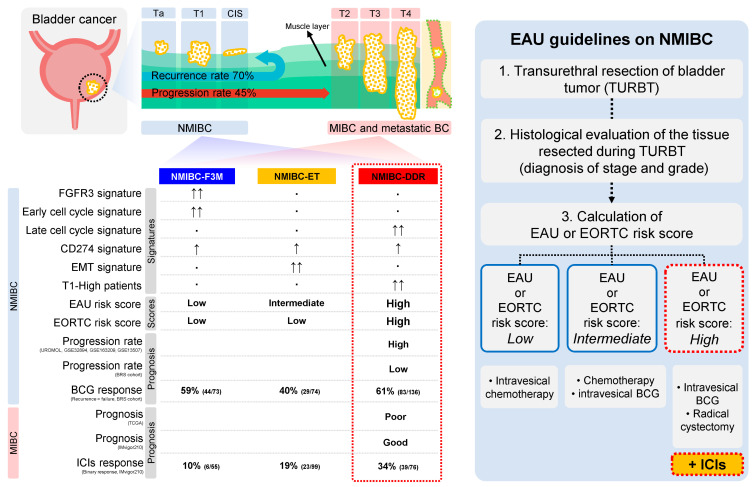
Summary of the characteristics of each class and the proposed immunotherapy for high-risk NMIBC patients. The multi-gene signature obtained from transcriptome data of NMIBC patients identified three distinct classes. Although the NMIBC-DDR class is considered to be high-risk NMIBC patients, they also respond best to immunotherapy such as BCG or ICIs. The number of arrows indicates relative activity.

**Table 1 ijms-25-03800-t001:** Baseline characteristics of the NMIBC and MIBC cohorts.

Variable	UROMOL2016	GSE32894	GSE163209	GSE13507	BRS	TCGA	IMvigor210
Total	476 (100)	213 (100)	217 (100)	102 (100)	283 (100)	407 (100)	298 (100)
Sex							
Male	367 (77.1)	158 (74.2)	159 (73.3)	86 (84.3)	227 (80.2)	301 (73.9)	233 (78.2)
Female	109 (22.9)	55 (25.8)	58 (26.7)	16 (15.7)	56 (19.8)	106 (26.1)	65 (21.8)
Grade							
NA	0	1 (0.5)	0	0	0	3 (0.1)	298 (100)
PUNLMP	7 (1.5)	0	0	0	0	0	0
Low or G1 and G2	277 (58.2)	139 (65.3)	77 (35.5)	85 (83.3)	15 (5.3)	21 (5.8)	0
High or G3 and G4	192 (40.3)	73 (34.2)	140 (64.5)	17 (16.7)	268 (94.7)	383 (94.1)	0
Stage							
CIS	3 (0.6)	0	0	0	0	0	0
Ta	345 (72.5)	116 (54.5)	113 (52.1)	23 (22.5)	38 (13.4)	0	0
T1	112 (23.5)	97 (45.5)	104 (47.9)	79 (77.5)	245 (86.6)	0	0
T2-4	16 (3.4)	0	0	0	0	407 (100)	298 (100)
Progression							
No	445 (93.5)	191 (89.7)	200 (92.2)	91 (89.2)	187 (66.1)	NA	NA
Yes	31 (6.5)	22 (10.3)	17 (7.8)	11 (10.8)	96 (33.9)	NA	NA
Death							
No	NA	NA	NA	NA	NA	229 (56.3)	110 (36.9)
Yes	NA	NA	NA	NA	NA	178 (43.7)	188 (63.1)
Treatment (BCG or ICIs)					BCG		ICIs
No	NA	NA	NA	NA	0	NA	0
Yes	NA	NA	NA	NA	283 (100)	NA	298 (100)

Abbreviations: PUNLMP, papillary urothelial neoplasm of low malignant potential; CIS, carcinoma in situ; BCG, Bacillus Calmette–Guérin; ICIs, immune checkpoint inhibitors; NA, not available.

**Table 2 ijms-25-03800-t002:** Univariate and multivariate Cox proportional hazards regression analysis in the UROMOL, BRS, and IMvigor210 cohorts.

Variable Disease Progression in the UROMOL Cohort	Univariate				Multivariate			
	hazardRatio	ciLower	ciUpper	*p*-Value	hazardRatio	ciLower	ciUpper	*p*-Value
Gender								
Male (Ref.) vs. Female	1.20	0.54	2.68	0.660				
Age								
≤69 (Ref.) vs. >69	2.49	1.14	5.40	0.021				
Tumor size								
≤3 cm (Ref.) vs. >3 cm	1.03	0.50	2.13	0.930				
Stage								
Ta (Ref.) vs. T1	9.11	4.19	19.81	<0.001				
Grade								
Low (Ref.) vs. High	4.31	1.98	9.38	<0.001				
Risk scores								
EAU Low (Ref.) vs. EAU High	2.13	1.01	4.48	0.046				
EORTC Low (Ref.) vs. EORTC High	12.47	4.36	35.66	<0.001	8.43	2.86	24.81	<0.001
Taxonomy								
Class 1, Class 2b, and Class 3 (Ref.) vs. Class 2a	5.06	2.38	10.77	<0.001				
The multi-gene signature								
Other classes (Ref.) vs. NMIBC-DDR	2.99	1.45	6.17	0.003	3.48	1.46	8.29	0.004
Variable Disease progression in the BRS cohort	Univariate				Multivariate			
	hazardRatio	ciLower	ciUpper	*p*-value	hazardRatio	ciLower	ciUpper	*p*-value
Gender								
Male (Ref.) vs. Female	0.46	0.24	0.87	0.018				
Age								
≤69 (Ref.) vs. >69	1.89	1.26	2.85	0.002				
Stage								
Ta (Ref.) vs. T1	1.11	0.61	1.99	0.727				
Grade								
Low (Ref.) vs. High	0.83	0.36	1.91	0.669				
Risk scores								
EORTC High (Ref.) vs. EORTC Highest	1.67	1.08	2.55	0.018	1.68	1.10	2.58	0.01
Taxonomy								
BRS2 and BRS3 (Ref.) vs. BRS1	0.55	0.34	0.90	0.017				
The multi-gene signature								
Other classes (Ref.) vs. NMIBC-DDR	0.63	0.41	0.97	0.037	0.62	0.04	0.95	0.03
Variable Overall survival in the IMvigor210 cohort	Univariate				Multivariate			
	hazardRatio	ciLower	ciUpper	*p*-value	hazardRatio	ciLower	ciUpper	*p*-value
Gender								
Male (Ref.) vs. Female	1.20	0.85	1.68	0.294				
PD-L1 expression in immune cell								
IC0 and IC1 (Ref.) vs. IC2	0.57	0.41	0.78	<0.001	0.56	0.41	0.78	<0.001
PD-L1 expression in tumor cell								
TC0 and TC1 (Ref.) vs. TC2	0.98	0.65	1.46	0.904				
Tumor mutation burden								
TMB Low and Intermediate (Ref.) vs. TMB High	0.52	0.36	0.74	<0.001				
The multi-gene signature								
Other classes (Ref.) vs. NMIBC-DDR	0.69	0.51	0.94	0.019	0.69	0.51	0.94	0.018

Abbreviations: EAU, European Association of Urology; EORTC, European Organization for Research and Treatment of Cancer; IC, PD-L1 expression in immune cells (IC with number indicates relative expression value); TC, PD-L1 expression in tumor cells (TC with number indicates relative expression value); TMB, tumor mutation burden.

## Data Availability

The gene expression data and clinical information are available in the GEO public database (https://www.ncbi.nlm.nih.gov/geo) under accession number GSE32894 and GSE163209. The gene expression data and clinical information of TCGA are available in the cancer browser (https://xenabrowser.net). The FASTq files and clinical information are available in the EGA public database (https://web2.ega-archive.org/) under accession numbers EGAS00001001236 (UROMOL2016), EGAS0001004693 (UROMOL2021), and EGAS00001002556 (IMvigor210 trial).

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
