# Peer review of "A Multi-Gene Signature of Non-Muscle-Invasive Bladder Cancer Identifies Patients Who Respond to Immunotherapies Including Bacillus Calmette–Guérin and Immune Checkpoint Inhibitors"

_ijms, 2024, doi:10.3390/ijms25073800_

Round 1
Reviewer 1 Report (New Reviewer)
Comments and Suggestions for Authors
Bladder cancer is a very common malignancy and approximately 75% of bladder cancer cases originate as non-muscle invasive bladder cancer (NMIBC). The recurrence rate NMIBC is high and it may advance to muscle-invasive bladder cancer (MIBC) and metastatic disease. BCG treatment is the standard adjuvant therapy for high-risk NMIBC patients after transurethral resection of bladder tumor, but with several limitations including the most important, ongoing production shortages. Therefore, alternative effective and readily available first-line treatments are need of the hour. Immunotherapy treatment may be a good alternative to overcome the BCG shortage. In the current study, the authors used transcriptomic data from NMIBC patients to construct a multi-gene signature to identify three molecularly distinct subtypes and did a validation analysis on six independent bladder cancer patient cohorts, including a cohort with BCG and immune check-point inhibitor (ICI) records, and estimated the prognostic relevance.
The authors accurately identified a gap is knowledge and addressed it. The work is very important and relevant. The manuscript is well written and I don’t have any concern.
Author Response
- Thank you so much for your kind words and recognition of our work. We truly appreciate your positive feedback and encouragement. Your acknowledgment of the importance and relevance of our research means a lot to us.
Reviewer 2 Report (New Reviewer)
Comments and Suggestions for Authors
Using a database of MNIBC patients, the authors identified different subtypes of NMIBC, NMIBC-DDR, NMIBC-F3M, and NMIBC-ET, and found that NMIBC-DDR has the highest rate of disease progression in NMIBC patients and is effective with BCG and ICI; the multi-gene signature identified high-risk NMIBC patients and may have a role in improving ICI responsiveness.
This study may be of value to the development of bioinformatics analysis of NMIBC. However, there are some points that should be reconsidered.
The manuscript does not include supplemental figures and tables and is difficult to understand.
The title is “a multi-gene signature identifies patients with high non-muscle-invasive bladder cancer who respond to immunotherapy”. The authors examined response to BCG in the BRS cohort of patients with NMIBC, while response to ICIs was examined in patients with MIBC. Therefore, this statement should be reconsidered.
The authors stated that the multi-gene signature may have a role in identifying high-risk NMIBC patients and improving the responsiveness of ICIs. In a clinical setting, should the expression of these many genes be examined in each patient before a setting a treatment? The practical clinical usefulness of this information also needs to be explained.
Table 1: Unclear what "NA" means, especially in the Grade column. An explanation of the abbreviation needs to be added underlined in the table.
Page 3, lines 92-98: The authors named clusters 2 and 3 NMIBC-F3M and NMIBC-ET, respectively. However, in the following sections, NMIBC-DDR, NMIBC-F3M, and NMIBC-ET are described in that order. For the reader's understanding, the order of the number of clusters 2 and 3 can be changed.
Page 5, line 129: Have the authors identified the five subtypes of NMIBC?
Page 5, line 153: Figure 4F should be indicated in the text.
Figure 4 A,B,C,E: Figure 4E shows that BCG improved the progression-free survival of NMIBC-DDR. However, the survival probabilities of NMIBC-ET and NMIBC-F3M in Figure 4E appear to be too low compared to those in Figures 4A,B,C. The lower probability for these types in Figure 4E appears to lead a better prognosis for NMIBC-DDR. This needs to be explained.
Figure 5A,B: Even if the cohorts of MIBC patients are different, the survival probabilities of NMIBC-ET and NMIBC-F3M in Figure 5B at 25 months are too low compared to those in Figure 5A. This discrepancy needs to be explained.
Table 2, Overall Survival in the Mvigor210 Cohort: The authors are investigating PD-L1 expression in immune cells. What is meant by increased PD-L1 expression in immune cells needs to be explained in more detail with references. Does PD-1 expression in immune cells, especially CD8+ T cells, play a role in patient survival? Also, the abbreviations TC0, TC1, TC2, and TMB need to be explained outside the underlining in the table.
Recent papers related to PD-1/PD-L1 in bladder cancer need to be cited. For example,
Li-P et al. identification of an immune-related risk signature correlates with immunophenotype and predicts anti-PD-1 efficacy of urothelial cancer. Front Cell Dev Biol. 2021;9:646982. doi: 10.3389/fcell.2021.646982.
Yan-Y et al. Identification of a novel immune microenvironment signature predicting survival and therapeutic options for bladder cancer. Aging (Albany NY). 2020;13(2):2780-2802. doi: 10.18632/aging.202327.
Chen-X et al. Cd8+ T effector and immune checkpoint signatures predict and responsiveness to immunotherapy in bladder cancer. Oncogene. 2021;40(43):6223-6234. doi: 10.1038/s41388-021-02019-6.
Comments on the Quality of English Language
can be improved.
Author Response
Using a database of MNIBC patients, the authors identified different subtypes of NMIBC, NMIBC-DDR, NMIBC-F3M, and NMIBC-ET, and found that NMIBC-DDR has the highest rate of disease progression in NMIBC patients and is effective with BCG and ICI; the multi-gene signature identified high-risk NMIBC patients and may have a role in improving ICI responsiveness. This study may be of value to the development of bioinformatics analysis of NMIBC. However, there are some points that should be reconsidered. The manuscript does not include supplemental figures and tables and is difficult to understand.
The title is “a multi-gene signature identifies patients with high non-muscle-invasive bladder cancer who respond to immunotherapy”. The authors examined response to BCG in the BRS cohort of patients with NMIBC, while response to ICIs was examined in patients with MIBC. Therefore, this statement should be reconsidered.
- Thank you for bringing this concern to our attention. Indeed, we acknowledge the distinction between the cohorts examined in our study regarding the response to immunotherapy. Our focus was to identify high-risk NMIBC patients who respond positively to immunotherapy such as both BCG and ICIs. Based on your suggestions, we have modified the title to better reflect patients with both NMIBC and MIBC.
- The revised title now reads: “A multi-gene signature of non-muscle-invasive bladder cancer identifies patients who respond to immunotherapies including Bacillus Calmette-Guérin and immune checkpoint inhibitors”
The authors stated that the multi-gene signature may have a role in identifying high-risk NMIBC patients and improving the responsiveness of ICIs. In a clinical setting, should the expression of these many genes be examined in each patient before a setting a treatment? The practical clinical usefulness of this information also needs to be explained.
- We completely agree with the importance of elucidating the practical clinical usefulness of our research findings. In addition, further investigation is needed to propose multigene signatures as clinically usable tools. We plan to select key genes that can effectively stratify patients in a clinical setting based on experimental follow-up studies. The results of this refined analysis could potentially be incorporated into a scoring system or diagnostic panel for evaluating high-risk NMIBC patients and predicting response to ICIs.
Table 1:
Unclear what "NA" means, especially in the Grade column. An explanation of the abbreviation needs to be added underlined in the table.
- Thanks to your detailed observations, we were able to make up for the shortcomings. The meanings of all abbreviations that the reader might be curious about, including the meaning of NA in the "Grade" line, are listed below the table.
Page 3, lines 92-98:
The authors named clusters 2 and 3 NMIBC-F3M and NMIBC-ET, respectively. However, in the following sections, NMIBC-DDR, NMIBC-F3M, and NMIBC-ET are described in that order. For the reader's understanding, the order of the number of clusters 2 and 3 can be changed.
- Thanks to your detailed observations, we were able to correct the confusing markings that were different from the supplementary figure1.
Page 5, line 129: Have the authors identified the five subtypes of NMIBC?
- Thanks to your careful observations, we were able to correct the number that were confusingly written different from the main text.
Page 5, line 153: Figure 4F should be indicated in the text.
- Thanks to your detailed observations, we were able to add figure numbers that were not indicated in the text.
Figure 4 A,B,C,E:
Figure 4E shows that BCG improved the progression-free survival of NMIBC-DDR. However, the survival probabilities of NMIBC-ET and NMIBC-F3M in Figure 4E appear to be too low compared to those in Figures 4A,B,C. The lower probability for these types in Figure 4E appears to lead a better prognosis for NMIBC-DDR. This needs to be explained.
Figure 5A,B:
Even if the cohorts of MIBC patients are different, the survival probabilities of NMIBC-ET and NMIBC-F3M in Figure 5B at 25 months are too low compared to those in Figure 5A. This discrepancy needs to be explained.
- We appreciate your insightful observation and the opportunity to clarify the interpretation of Figure 4E in comparison to Figures 4A, B, and C. Figure 4E displayed the survival analysis results from the BRS cohort, predominantly consisting of patients with high-grade NMIBC, as indicated in Table 1. Therefore, it is important to note that the overall prognosis of patients in the BRS cohort may not be as favorable compared to other NMIBC cohorts.
- In Figures 5A and B, the same 25-month survival probabilities are also lower for patients in the IMvigor210 cohort. This is because, unlike the TCGA cohort comprised of patients with muscle-invasive bladder cancer (MIBC), the patients in the IMvigor210 cohort were clinical trials targeting patients with metastatic muscle-invasive bladder cancer (metastatic MIBC or advanced urothelial carcinoma). Therefore, even during the same period, patients in the IMvigor210 cohort have a relatively poor prognosis.
- In response to your feedback, we have revised the discussion section to explicitly address these points and provide further context for the interpretation. We hope this clarification helps to enhance the understanding of our findings.
- The revised discussion section now reads: Because the BRS cohort mainly included patients with high-grade NMIBC (94.7%, Table 1) and the IMvigor210 cohort was a clinical trial conducted on patients with metastatic muscle-invasive bladder cancer [ref], In this study, the progression-free survival of the BRS cohort or the overall survival of the IMvigor210 cohort of MIBC may show a much worse trend than other compared cohorts.
Table 2, Overall Survival in the Mvigor210 Cohort:
The authors are investigating PD-L1 expression in immune cells. What is meant by increased PD-L1 expression in immune cells needs to be explained in more detail with references. Does PD-1 expression in immune cells, especially CD8+ T cells, play a role in patient survival? Also, the abbreviations TC0, TC1, TC2, and TMB need to be explained outside the underlining in the table.
Recent papers related to PD-1/PD-L1 in bladder cancer need to be cited. For example,
Li-P et al. identification of an immune-related risk signature correlates with immunophenotype and predicts anti-PD-1 efficacy of urothelial cancer. Front Cell Dev Biol. 2021;9:646982. doi: 10.3389/fcell.2021.646982.
Yan-Y et al. Identification of a novel immune microenvironment signature predicting survival and therapeutic options for bladder cancer. Aging (Albany NY). 2020;13(2):2780-2802. doi: 10.18632/aging.202327.
Chen-X et al. Cd8+ T effector and immune checkpoint signatures predict and responsiveness to immunotherapy in bladder cancer. Oncogene. 2021;40(43):6223-6234. doi: 10.1038/s41388-021-02019-6.
- The effect of increased PD-L1 expression on immune cells and especially CD8-T cells on patient survival was explained with reference to three papers. Additionally, abbreviations have been added to the bottom of Table 2.
- The revised discussion section now reads: PD-L1 is expressed on tumor cells and peripheral immune cells and is an important part of tumor immune evasion mechanisms. Many follow-up studies have revealed that PD-L1 expression in immune cells is a major target for immunotherapy [20, 21]. PD-L1 expression, especially on CD8-T cells, may be involved in tumor evasion of immune re-sponses. Excessive PD-L1 expression is associated with impaired function of CD8-T cells, which may have negative effects on tumor invasion and metastasis [22].
We extend our sincere gratitude for your thoughtful review and insightful comments. Your dedication of time and expertise is truly appreciated, and we are confident that your constructive feedback has significantly enhanced the quality of our manuscript.

This manuscript is a resubmission of an earlier submission. The following is a list of the peer review reports and author responses from that submission.
Round 1
Reviewer 1 Report
Comments and Suggestions for Authors
The study presents a comprehensive analysis of NMIBC patient data to identify five molecularly distinct subtypes that could potentially benefit from immunotherapy. It demonstrates the use of a multi-gene signature to predict patient prognosis. They found that patients in the DNA damage repair group had the highest rates of disease progression but also benefited most from immune checkpoint inhibitors (ICIs). The study suggests that the multi-gene signature could help identify high-risk NMIBC patients and improve the responsiveness of ICIs. The researchers propose immunotherapy as a new first-line treatment in light of the BCG shortage. However, one major and several other limitations and areas require further clarification:
1. The study uses data from public databases. Do data collection methods, patient demographics, and treatment protocols differ across different studies? The authors should clarify how they accounted for these potential biases in their analysis. For example, there are slight differences between the selection criteria of the clinical trials of IMvigor010 and IMvigor210 and there is no ICI-responsive group in 010, but DDR responds in 210. 010 accepts patients with a high risk of recurrence, and 210 accepts patients with advanced cancer. Unless the authors can explain how the difference in cohorts can result in such a difference in drug response, it is challenging to suggest that ICI improves the outcome since there is no responsive group in 010 (especially when the N is imbalanced for groups DDR and Fgfr3). Although not significant, the Imvigor010 should also be included as a row in Figure 6. The authors should also clarify how they handled missing data in their analysis, if there was any, as this could potentially bias the results.
2. Why do authors exclude other IMvigor trials? For example IMvigor110
3. Do authors consider the negative Z-scores when stratifying or predicting the prognosis, or is it only increased expression? If they do not, why not?
4. In vitro or in vivo experiments are needed to confirm the findings and elucidate the underlying mechanisms. The real-world applicability of the multi-gene signature needs to be explored. This aspect should be mentioned in the manuscript.
5. The use of machine learning algorithms to analyze the data, while innovative, could also introduce bias or overfitting. the authors should also explain how the hyperparameter strategy prevents overfitting. Also, the explanation of the details of the machine learning algorithms used is not sufficient for the replication and validation of the results. The authors should provide more information about the randomForest. I see that the authors included their parameters and performance metrics in the supplementary for the evaluation of the output, however, the methodology itself should be explained in more detail.
6. The study does not discuss potential confounding factors that could influence the results, such as patient age, comorbidities, and lifestyle factors. The authors should discuss how/if they controlled for these factors in their analysis.
7. The lines between 204 & 214 belong to the methods section.
Overall, while the study provides insights into the potential benefits of immunotherapy for NMIBC patients, authors need to address the limitations of the work more rigorously to validate these findings and explore their implications for clinical practice. Also, I wonder if the findings would be more significant when the patients are stratified into 2 groups based on drug response.
Author Response
Reviewer #1
Comments and Suggestions for Authors
The study presents a comprehensive analysis of NMIBC patient data to identify five molecularly distinct subtypes that could potentially benefit from immunotherapy. It demonstrates the use of a multi-gene signature to predict patient prognosis. They found that patients in the DNA damage repair group had the highest rates of disease progression but also benefited most from immune checkpoint inhibitors (ICIs). The study suggests that the multi-gene signature could help identify high-risk NMIBC patients and improve the responsiveness of ICIs. The researchers propose immunotherapy as a new first-line treatment in light of the BCG shortage. However, one major and several other limitations and areas require further clarification:
1. The study uses data from public databases. Do data collection methods, patient demographics, and treatment protocols differ across different studies? The authors should clarify how they accounted for these potential biases in their analysis. For example, there are slight differences between the selection criteria of the clinical trials of IMvigor010 and IMvigor210 and there is no ICI-responsive group in 010, but DDR responds in 210. 010 accepts patients with a high risk of recurrence, and 210 accepts patients with advanced cancer. Unless the authors can explain how the difference in cohorts can result in such a difference in drug response, it is challenging to suggest that ICI improves the outcome since there is no responsive group in 010 (especially when the N is imbalanced for groups DDR and Fgfr3). Although not significant, the Imvigor010 should also be included as a row in Figure 6. The authors should also clarify how they handled missing data in their analysis, if there was any, as this could potentially bias the results.
2. Why do authors exclude other IMvigor trials? For example, IMvigor110
> To reduce as much as possible potential bias in the published data used in this study, we feel it necessary to clearly present to the reader the differences between the different trials. Additionally, the IMvigor210 trial met the trial's endpoints, while the IMvigor010 trial did not. We added these details to Figure 6 and modified the discussion section in the manuscript so that readers can fully understand the results of this study. (Revised Figure 6, modified Discussion section, 243-253 lines)
3. Do authors consider the negative Z-scores when stratifying or predicting the prognosis, or is it only increased expression? If they do not, why not?
> Both positive and negative Z-scores were considered when stratifying or predicting prognosis. Additionally, the gene expression data used to build the prediction model was preprocessed through normalization and normalization. However, since these were not included in the manuscript, we added them to the methods section (Materials and Methods, 4.3 section, 312-313 lines).
4. In vitro or in vivo experiments are needed to confirm the findings and elucidate the underlying mechanisms. The real-world applicability of the multi-gene signature needs to be explored. This aspect should be mentioned in the manuscript.
> We believe that experiments is essential to confirm and validate research results. Limitations of these inclusions are noted in the discussion section (Discussion, 260-267 lines).
5. The use of machine learning algorithms to analyze the data, while innovative, could also introduce bias or overfitting. the authors should also explain how the hyperparameter strategy prevents overfitting. Also, the explanation of the details of the machine learning algorithms used is not sufficient for the replication and validation of the results. The authors should provide more information about the randomForest. I see that the authors included their parameters and performance metrics in the supplementary for the evaluation of the output, however, the methodology itself should be explained in more detail.
> We completely agree that the definition of the hyper-parameters (mtry, node size, and sample size) used to construct the prediction model are lacking and that they should be sufficiently explained to the reader to prevent overfitting. Therefore, we have tried to provide additional information in the Materials and Methods section to help readers fully understand it (Materials and Methods, 4.3 section, 308-313 lines).
6. The study does not discuss potential confounding factors that could influence the results, such as patient age, comorbidities, and lifestyle factors. The authors should discuss how/if they controlled for these factors in their analysis.
> We fully understand your advice. We recognized a need to provide readers with information to evaluate how multi-gene signatures perform in comparison to existing clinical information. Cox hazard regression analysis was performed to compare and analyze how clinicopathological factors (such as age, gender, stage, and grade) of the cohorts used in this study may affect prognosis. (Result, 2.4 section)
7. The lines between 204 & 214 belong to the methods section.
> We fully understand your advice and have included it in the Materials and Methods section (300-304 lines).
Overall, while the study provides insights into the potential benefits of immunotherapy for NMIBC patients, authors need to address the limitations of the work more rigorously to validate these findings and explore their implications for clinical practice. Also, I wonder if the findings would be more significant when the patients are stratified into 2 groups based on drug response.
> We did not perform the analysis in that manner because we were concerned that there might be problems dividing patients into two groups based on drug response. Clearly, separating them into two groups would yield stronger results. However, the data we used in our analysis are characterized by a lot of information. An example is tissue from a cancer patient obtained through biopsy, which may contain various types of cells, including cancer cells as well as the tumor microenvironment. Considering the characteristics of these data, the method you suggest may produce more positive results when using single cell transcriptome data.

Reviewer 2 Report
Comments and Suggestions for Authors
The manuscript entitled "A multi-gene signature identifies patients with high-risk non-muscle-invasive bladder cancer who respond to immunotherapy" submitted to the journal International Journal of Molecular Sciences with manuscript ID: ijms-2863550 explored a multi-gene signature in high-risk independent non-muscle-invasive bladder cancer (NMIBC) cohorts to evaluate the benefit of immune checkpoint inhibitors usage.
Authors stratified NMIBC patients into five molecularly distinct subtypes (metabolic, FGFR3, 211 EMT+Tcell, DDR+Tcell, and DDR) and suggested that cases from the DDR class may have potential benefits of immunotherapy.
The manuscript is well-written and provides new knowledge in the studied field.
I have just a few minor concerns about the heterogeneity of used cohorts.
Regression analyses of selected clinical features and molecular subtypes should be added.
The genetic heterogeneity also could affect the obtained data and should be considered as a limitation.
I hope my review is helpful.
Author Response
Reviewer #2
Comments and Suggestions for Authors
The manuscript entitled "A multi-gene signature identifies patients with high-risk non-muscle-invasive bladder cancer who respond to immunotherapy" submitted to the journal International Journal of Molecular Sciences with manuscript ID: ijms-2863550 explored a multi-gene signature in high-risk independent non-muscle-invasive bladder cancer (NMIBC) cohorts to evaluate the benefit of immune checkpoint inhibitors usage.
Authors stratified NMIBC patients into five molecularly distinct subtypes (metabolic, FGFR3, 211 EMT+Tcell, DDR+Tcell, and DDR) and suggested that cases from the DDR class may have potential benefits of immunotherapy.
The manuscript is well-written and provides new knowledge in the studied field.
I have just a few minor concerns about the heterogeneity of used cohorts.
Regression analyses of selected clinical features and molecular subtypes should be added.
> We fully understand your advice. We recognized a need to provide readers with information to evaluate how multi-gene signatures perform in comparison to existing clinical information. Cox hazard regression analysis was performed to compare and analyze how clinicopathological factors (such as age, gender, stage, and grade) of the cohorts used in this study may affect prognosis. (Result, 2.4 section)
The genetic heterogeneity also could affect the obtained data and should be considered as a limitation.
> To reduce as much genetic heterogeneity as possible in the published data used in this study, we believe it is necessary to clearly present the differences between the different trials to the reader. For example, the IMvigor210 trial met its endpoints, whereas the IMvigor010 trial did not. To help readers fully understand the results of this study, we added these details to Figure 6 and modified the discussion section in the manuscript (Modified Figure 6, Modified Discussion Section).
I hope my review is helpful.

Round 2
Reviewer 1 Report
Comments and Suggestions for Authors
1. The study uses data from public databases. Do data collection methods, patient demographics, and treatment protocols differ across different studies? The authors should clarify how they accounted for these potential biases in their analysis. For example, there are slight differences between the selection criteria of the clinical trials of IMvigor010 and IMvigor210 and there is no ICI-responsive group in 010, but DDR responds in 210. 010 accepts patients with a high risk of recurrence, and 210 accepts patients with advanced cancer. Unless the authors can explain how the difference in cohorts can result in such a difference in drug response, it is challenging to suggest that ICI improves the outcome since there is no responsive group in 010 (especially when the N is imbalanced for groups DDR and Fgfr3). Although not significant, the Imvigor010 should also be included as a row in Figure 6. The authors should also clarify how they handled missing data in their analysis, if there was any, as this could potentially bias the results.
2. Why do authors exclude other IMvigor trials? For example, IMvigor110
> To reduce as much as possible potential bias in the published data used in this study, we feel it necessary to clearly present to the reader the differences between the different trials. Additionally, the IMvigor210 trial met the trial's endpoints, while the IMvigor010 trial did not. We added these details to Figure 6 and modified the discussion section in the manuscript so that readers can fully understand the results of this study. (Revised Figure 6, modified Discussion section, 243-253 lines)
Thanks for the responses. My original comment about the major concern was two-pronged, please let me elaborate.
a) Why do authors exclude other IMvigor trials such as IMvigor 110? There are multiple other Atezolizumab trials on NMIBC. What are the authors' inclusion-exclusion criteria for this data?
b) Can the authors explain how the difference in cohorts can result in such a difference in drug response?
Please elaborate on the results by postulating how the selection criteria can result in such a difference in ICI response instead of only stating that the selection criteria are different. Please postulate on possibilities.
Additionally, now that Fig 6 is updated, the results show that all groups were similarly responsive to ICI in the IMvigor010 trial, and the least responsive group was 46%, which is considered good. The authors should include a discussion about these new findings.
More importantly, with the updated Fig 6, I think the study suffers heavily from statistical bias now. IMvigor010 clinical trial did not meet its primary endpoint of improved disease-free survival in the atezolizumab group over observation, IMvigor210 did. Such an outcome makes the prediction based on the multi-gene signatures more crucial for IMvigor010 data, doesn't it? Because if multi-gene signature-based prediction helped predict the drug response, we should have seen that help in the failed trial. On the other hand, the IMvigor210 had already succeded in satisfying the endpoint as the authors noted without the help of the authors' prediction method. So a reader can conclude that the multi-gene signatures are not useful in identifying patients who can benefit from ICI treatment. In other words, the results are biased toward the success of the clinical trial data that they are applied to. If there is no logical flaw in this type of conclusion, and unless the authors can reject it, the study cannot be suited for publication.
Author Response
Dear Reviewer,
We express our gratitude for your diligent review of our manuscript titled "A Multi[1]Gene Signature Identifies Patients with High-Risk Non-Muscle-Invasive Bladder Cancer Who Respond to Immunotherapy" (Manuscript ID Ijms-2863550). We highly appreciate the thorough critiques and constructive feedback provided by the reviewers.
In response to their comments, we have invested considerable effort in revising the manuscript. The raised questions have been meticulously addressed, and the requested information has been incorporated. We sincerely value these insightful comments, and we believe that the revised manuscript has been significantly enhanced, showcasing the increased clarity and significance of our findings. A comprehensive point-by-point response to the reviewers’ comments is enclosed, with changes to the original manuscript indicated by blue-colored in-text.
We appreciate your time and dedication to this review process. We eagerly anticipate your decision.
Sincerely yours,
The details of the revisions
Reviewer#1
- Figure 6; Discussion section, 243-253 lines
- Figure 6; Discussion section, 243-253 lines
- Materials and Methods, 4.3 section, 312-313 lines
- Discussion, 260-267 lines
- Materials and Methods, 4.3 section, 308-313 lines
- Result, 2.4 section
- Materials and Methods, 4.3 section, 300-304 lines
Reviewer #1
Comments and Suggestions for Authors
The study presents a comprehensive analysis of NMIBC patient data to identify five molecularly distinct subtypes that could potentially benefit from immunotherapy. It demonstrates the use of a multi-gene signature to predict patient prognosis. They found that patients in the DNA damage repair group had the highest rates of disease progression but also benefited most from immune checkpoint inhibitors (ICIs). The study suggests that the multi-gene signature could help identify high-risk NMIBC patients and improve the responsiveness of ICIs. The researchers propose immunotherapy as a new first-line treatment in light of the BCG shortage. However, one major and several other limitations and areas require further clarification:
- The study uses data from public databases. Do data collection methods, patient demographics, and treatment protocols differ across different studies? The authors should clarify how they accounted for these potential biases in their analysis. For example, there are slight differences between the selection criteria of the clinical trials of IMvigor010 and IMvigor210 and there is no ICI-responsive group in 010, but DDR responds in 210. 010 accepts patients with a high risk of recurrence, and 210 accepts patients with advanced cancer. Unless the authors can explain how the difference in cohorts can result in such a difference in drug response, it is challenging to suggest that ICI improves the outcome since there is no responsive group in 010 (especially when the N is imbalanced for groups DDR and Fgfr3). Although not significant, the Imvigor010 should also be included as a row in Figure 6. The authors should also clarify how they handled missing data in their analysis, if there was any, as this could potentially bias the results.
- Why do authors exclude other IMvigor trials? For example, IMvigor110>
-> To reduce as much as possible potential bias in the published data used in this study, we feel it necessary to clearly present to the reader the differences between the different trials. Additionally, the IMvigor210 trial met the trial's endpoints, while the IMvigor010 trial did not. We added these details to Figure 6 and modified the discussion section in the manuscript so that readers can fully understand the results of this study. (Revised Figure 6, modified Discussion section, 243-253 lines)
- Do authors consider the negative Z-scores when stratifying or predicting the prognosis, or is it only increased expression? If they do not, why not?
-> Both positive and negative Z-scores were considered when stratifying or predicting prognosis. Additionally, the gene expression data used to build the prediction model was preprocessed through normalization and normalization. However, since these were notincluded in the manuscript, we added them to the methods section (Materials and Methods, 4.3 section, 312-313 lines).
- In vitro or in vivo experiments are needed to confirm the findings and elucidate the underlying mechanisms. The real-world applicability of the multi-gene signature needs to be explored. This aspect should be mentioned in the manuscript.
-> We believe that experiments is essential to confirm and validate research results. Limitations of these inclusions are noted in the discussion section (Discussion, 260-267 lines)
- The use of machine learning algorithms to analyze the data, while innovative, could also introduce bias or overfitting. the authors should also explain how the hyperparameter strategy prevents overfitting. Also, the explanation of the details of the machine learning algorithms used is not sufficient for the replication and validation of the results. The authors should provide more information about the randomForest. I see that the authors included their parameters and performance metrics in the supplementary for the evaluation of the output, however, the methodology itself should be explained in more detail.
-> We completely agree that the definition of the hyper-parameters (mtry, node size, and sample size) used to construct the prediction model are lacking and that they should be sufficiently explained to the reader to prevent overfitting. Therefore, we have tried to provide additional information in the Materials and Methods section to help readers fully understand it (Materials and Methods, 4.3 section, 308-313 lines).
- The study does not discuss potential confounding factors that could influence the results, such as patient age, comorbidities, and lifestyle factors. The authors should discuss how/if they controlled for these factors in their analysis.
-> We fully understand your advice. We recognized a need to provide readers with information to evaluate how multi-gene signatures perform in comparison to existing clinical information. Cox hazard regression analysis was performed to compare and analyze how clinicopathological factors (such as age, gender, stage, and grade) of the cohorts used in this study may affect prognosis. (Result, 2.4 section)
- The lines between 204 & 214 belong to the methods section.
-> We fully understand your advice and have included it in the Materials and Methods section(300-304 lines).
Overall, while the study provides insights into the potential benefits of immunotherapy for NMIBC patients, authors need to address the limitations of the work more rigorously to validate these findings and explore their implications for clinical practice. Also, I wonder if the findings would be more significant when the patients are stratified into 2 groups based on drug response.
-> We did not perform the analysis in that manner because we were concerned that there might be problems dividing patients into two groups based on drug response. Clearly, separating them into two groups would yield stronger results. However, the data we used in our analysis are characterized by a lot of information. An example is tissue from a cancer patient obtained through biopsy, which may contain various types of cells, including cancer cells as well as the tumor microenvironment. Considering the characteristics of these data, the method you suggest may produce more positive results when using single cell transcriptome data.

Round 3
Reviewer 1 Report
Comments and Suggestions for Authors
The authors believe that their method of classifying multi-gene signatures can help predict the ICI response in NMIBC. The authors argue that if the patients of a clinical trial had been selected based on multi-gene signatures, the clinical trial would have been successful. However, they proceed to apply their method to data from an already successful clinical trial instead of an already unsuccessful one as the basis of their argument.
The authors fail to design the research appropriately and support the conclusion that they have drawn. Therefore, the study does not merit publication at IJMS.